

# Multiple instance learning-based prediction of programmed death-ligand 1 (PD-L1) expression from hematoxylin and eosin (H&E)-stained histopathological images in breast cancer

Zhen Da[1,2,*], Heng Yang[3,4,*], Bianba Zhaxi[5], Kaixiang Sun[3,4], Guohui Bai[5], Chao Wang[6], Feiyan Wang[3,4], Weijun Pan[4,6] and Rui Du[1,2]

[1] Department of Pathology, People's Hospital of Xizang Autonomous Region, Lhasa, Xizang, China
[2] Department of Pathology, Guangdong Women and Children Hospital, Guangzhou, Guangdong, China
[3] Kunyuan Fangqing Medical Technology Co., LTD, Guangzhou, Guangdong, China
[4] Jinfeng Laboratory, Chongqing, China
[5] Department of General Surgery, People's Hospital of Xizang Autonomous Region, Lhasa, Xizang, China
[6] Department of Pathology, Nanfang Hospital and Basic Medical College, Southern Medical University, Guangzhou, Guangdong, China
[*] These authors contributed equally to this work.

Corresponding authors
Rui Du, gdsfydr@sina.com
Weijun Pan, weijunp@126.com

## ABSTRACT

Programmed death-ligand 1 (PD-L1) is an important biomarker increasingly used as a predictive marker in breast cancer immunotherapy. Immunohistochemical quantification remains the standard method for assessment. However, it presents challenges related to time, cost, and reliability. Hematoxylin and eosin (H&E) staining is a routine method in cancer pathology, known for its accessibility and consistently reliability. Deep learning has shown the potential in predicting biomarkers in cancer histopathology. This study employs a weakly supervised multiple instance learning (MIL) approach to predict PD-L1 expression from H&E-stained images using deep learning techniques. In the internal test set, the TransMIL method achieved an area under the curve (AUC) of 0.833, and in an independent external test set, it achieved an AUC of 0.799. Additionally, since RNA sequencing results indicate a threshold that allows for the separation of H&E pathology images, we further validated our approach using the public TCGA-TNBC dataset, achieving an AUC of 0.721. These findings demonstrates that the Transformer-based TransMIL model can effectively capture highly heterogeneous features within the MIL framework, exhibiting strong cross-center generalization capabilities. Our study highlights that appropriate deep learning techniques can enable effective PD-L1 prediction even with limited data, and across diverse regions and centers. This not only underscores the significant potential of deep learning in pathological artificial intelligence (AI) but also provides valuable insights for the rational and efficient allocation of medical resources.

## INTRODUCTION

Globally, breast cancer is one of the most common cancers among women, with its incidence increasing yearly (*Katsura et al., 2022*). Programmed cell death protein 1 (PD-1) and programmed cell death ligand 1 (PD-L1) are immunoregulatory proteins that play essential roles in the immune system (*Kornepati, Vadlamudi & Curiel, 2022*). PD-1 is a membrane receptor protein primarily expressed on the surface of immune cells such as T cells, B cells, NK cells, and certain dendritic cells. PD-L1, the ligand for PD-1, is primarily expressed on the surface of various cells, including tumor cells, immune cells, and certain normal tissue cells. When PD-1 binds to PD-L1, it inhibits T cell activation and function, thereby suppressing the immune response to prevent excessive immune reactions that could damage normal tissues. The interaction between PD-1 and PD-L1 is especially important within the tumor microenvironment (TME), as tumor cells can exploit PD-L1 to evade the immune system, enhancing their chances of survival and growth. Antibody therapies targeting PD-1 or PD-L1 have become vital strategies in cancer immunotherapy, and by blocking the PD-1 and PD-L1 interaction, T cell activation and function are restored, strengthening the immune system's attack on tumors for an antitumor effect (*Sun, Mezzadra & Schumacher, 2018*; *Wang, Dougan & Dougan, 2023*).

Based on these functions, PD-1/PD-L1 inhibitor immunotherapy has emerged as one of the most promising cancer treatment methods, providing patients with a personalized, minimally invasive treatment option. According to clinical trial reports, the Food and Drug Administration (FDA) has approved reagents such as SP263 and 22C3 as companion diagnostics for PD-L1, and PD-1/PD-L1 immunotherapy combinations approved by the FDA have shown positive results in cancers like lung cancer, colorectal cancer, and triple negative breast cancer (TNBC) (*Lee et al., 2020*; *Nobin et al., 2024*; *Savic et al., 2019*). Studies indicate that atezolizumab combined with nab-paclitaxel therapiesbenefits PD-L1 positive patients with unresectable locally advanced or metastatic TNBC (*Kwapisz, 2021*). However, only a subset of patients responds to immunotherapy. Immunohistochemistry (IHC) staining for PD-L1 is currently the standard assessment method for PD-L1, in which pathologists evaluate CPS/TPS/IC scores based on IHC-stained slides to determine positivity or negativity (*Yeong et al., 2022*). Numerous PD-L1 staining kits with varying standards are commercially available. In addition, high reagent costs, prolonged staining time, and manual interpretation errors can impact reliability, limiting access in regions with constrained medical or economic resources, and narrowing the range of patients who could benefit.

Since 2019, research has shown that deep learning can detect biomarkers directly from digitized hematoxylin and eosin (H&E) stained tissue slides (*Bergstrom et al., 2023*; *Lazard et al., 2022*; *Schirris & Horlings, 2022*; *Valieris et al., 2020*; *Zhang et al., 2023*). However, manual annotation of medical images is time-consuming, labor-intensive, and costly, leading to limited annotated data, especially for whole-slide images (WSI), where annotation is particularly scarce. To address this shortage of medical image annotations, weakly supervised methods have become a promising solution (*Laleh et al., 2022*; *Qian et al., 2022*; *Wang et al., 2019*). Multiple instance learning (MIL) is a specialized form of

weak supervision, where data is organized as "bags", each containing multiple "instances", allowing label prediction at the "bag" level (*Herrera et al., 2016*). These methods train models with minimal labeled data, effectively addressing data bottlenecks in medical image analysis, and have achieved some progress in pathology image analysis (*Carbonneau et al., 2018*).

Convolutional neural networks (CNNs) methods process multiple small patches to extract image features and use aggregation modules for classification, ultimately generating a single prediction for the patient. Recently, Transformer neural networks, known for their outstanding performance and robustness, have been introduced into pathology. Transformers are not only used for feature extraction but also perform exceptionally as aggregation models, providing new possibilities for pathology image analysis (*Chen et al., 2022*; *Shao et al., 2021*). This Transformer-based approach to weakly supervised feature extraction and aggregation is expected to enhance the model's analytical capability for pathology images under limited annotation conditions, advancing machine learning applications in medicine.

Therefore, to improve the efficiency, reduce the cost and improve the consistency of PD-L1 assays, this study employed a MIL-based detection method to predict PD-L1 expression in H&E-stained breast cancer WSIs from local multicenter datasets spanning regions of varying economic development levels. Results indicated a significant association between H&E-stained breast cancer slides and PD-L1 expression. For example, the best-performing model in this study achieved an area under the curve (AUC) of 0.833 in the dataset acquired from Nanfang Hospital (NF) and Huayin Health (HY) test sets, with an AUC of 0.762 for biopsy samples and 0.856 for surgical samples. Additionally, the model performed well across centers, achieving an AUC of 0.799 in the independent external test set acquired from People's Hospital of Xizang Autonomous Region (XZ). To enhance model interpretability, this study used heatmaps to display features of strongly and weakly correlated regions. Features such as lymphocyte infiltration and blood vessel formation in H&E-stained images were significantly associated with high PD-L1 expression. These findings demonstrate that tumor molecular characteristics can be correlated with tissue and cellular morphology through deep learning. Training with appropriate models can improve understanding of these associations and may significantly impact insights into tumor development and therapeutic strategy formulation.

## MATERIALS AND METHODS

The overall workflow of this study is shown as Fig. 1. We used the collected NF and HY datasets for training and testing our method to validate its effectiveness on cross-region datasets, and use the TCGA-TNBC dataset as an external validation set to demonstrate the generalization of our method. The whole method mainly consists of feature extraction, and MIL-based Patch aggregation to extract the overall features of large-size WSIs for classification. In the following, we describe each step in detail.
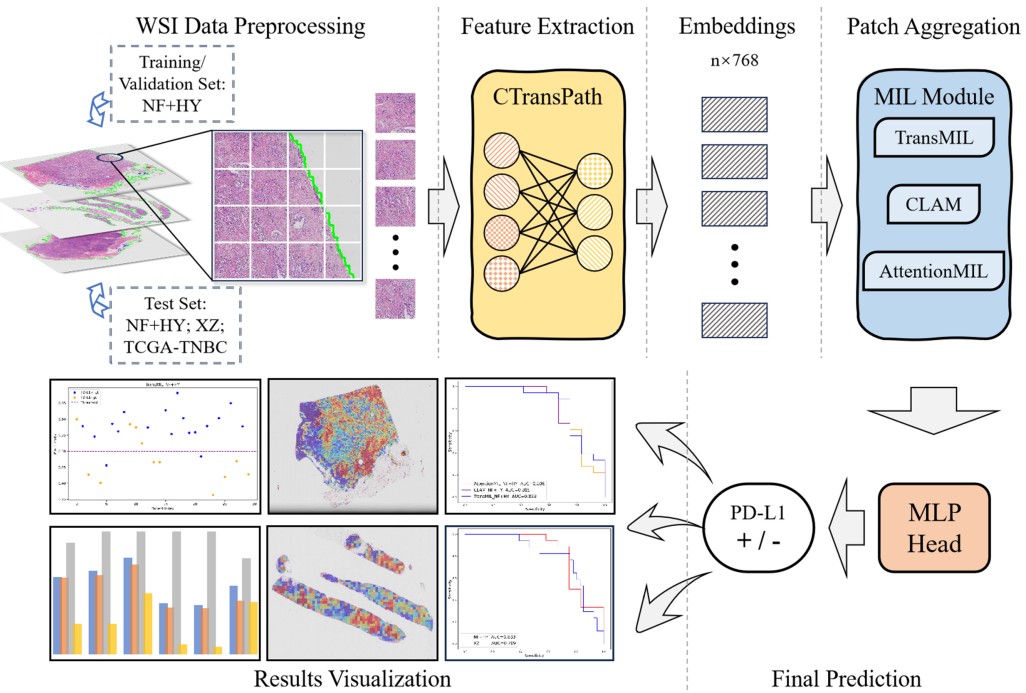

**Figure 1** **Workflow overview with image pre-processing, modeling, performance metrics.** The model training and inference workflow consists of three steps. First, patient data is divided into training, validation, and test sets, followed by image augmentation. Next, the CTransPath model, based on the Swin Transformer, is used as the feature extraction network to extract 768-dimensional features from each patch. Finally, three MIL-based weakly supervised aggregation modules are employed to aggregate the patch features and predict the PD-L1 expression levels at the WSI level. The final output includes receiver operating characteristic (ROC) curves and AUC values, visualized heatmaps, and statistical analysis images.

## Data acquisition and processing

This study was conducted on breast cancer histological samples from three independent cohorts: NF, HY, and XZ (Table 1). The NF cohort consists of 131 patients collected between 2020 and 2024, the HY cohort includes 56 patients collected between 2022 and 2023, and the XZ cohort represents an economically underdeveloped region, with 100 patients collected between 2017 and 2022. The study was approved by the Ethics Committee of People's Hospital of Xizang Autonomous Region (No. ME-TBHP-22-KJ-041) on July 22, 2023; a waiver of informed consent was granted by the same committee on July 4, 2022. Pathology experts reviewed all available H&E and PD-L1 IHC images and annotated the PD-L1 positive or negative status for all data in these three cohorts. During the annotation process, some patients were excluded from analysis for one of the following reasons: missing H&E or IHC images, non-breast tissue, non-primary diagnosis, or out-of-focus images. The PD-L1 status for the remaining patients was annotated as either negative or positive.

The IHC staining used FDA-approved PD-L1 antibodies (Dako 22C3; Ventana SP263) as companion diagnostics, with PD-L1 (22C3) CPS ≥ 1 and PD-L1 (SP263) TPS ≥ 1 as

**Table 1 Cohort data of each data group.** The total number of patients with inclusion and exclusion statistics for the three local centers are presented in this table.

| Biomarker | PD-L1 | | | | | |
|---|---|---|---|---|---|---|
| Cohort | NF (2020–2024) | | HY (2022–2023) | | XZ (2017–2022) | |
| **Total patients** | **n** | **%** | **n** | **%** | **n** | **%** |
| Total | 131 | 100.0% | 56 | 100.0% | 100 | 100.0% |
| Excluded from analysis | 22 | 16.8% | 1 | 1.8% | 42 | 42.0% |
| Included in analysis | 109 | 83.2% | 55 | 98.2% | 58 | 58.0% |
| **Patients excluded from analysis** | **n** | **%** | **n** | **%** | **n** | **%** |
| Total | 22 | 100.0% | 1 | 100.0% | 42 | 100.0% |
| Non-mammary tissue | 20 | 91.0% | 0 | 0.0% | 1 | 2.4% |
| Non-initial diagnosis | 1 | 4.5% | 0 | 0.0% | 1 | 2.4% |
| Out of focus | 1 | 4.5% | 1 | 100.0% | 0 | 0.0% |
| Deficient staining | 0 | 0.0% | 0 | 0.0% | 40 | 95.2% |
| **Patients included in analysis** | **n** | **%** | **n** | **%** | **n** | **%** |
| Total | 109 | 100.0% | 55 | 100.0% | 58 | 100.0% |
| Negative (CPS < 1 or TPS < 1) | 37 | 33.9% | 23 | 41.8% | 41 | 70.7% |
| Positive (CPS ≥ 1 or TPS ≥ 1) | 72 | 66.1% | 32 | 58.2% | 17 | 29.3% |
| Puncture | 53 | 48.6% | 16 | 29.1% | 23 | 39.7% |
| Surgery | 56 | 51.4% | 39 | 70.9% | 35 | 60.3% |
| TNBC | 11 | 10.1% | —— | —— | 8 | 13.8% |
| Training set | 60 | 55.0% | 33 | 60.0% | 0 | 0.0% |
| Validation set | 19 | 17.5% | 11 | 20.0% | 0 | 0 |
| Test set | 30 | 27.5% | 11 | 20.0% | 58 | 100.0% |

criteria for PD-L1 positivity. Each slide's PD-L1 status (positive or negative) was annotated based on the interpretation by clinical pathologists. All slides from the three cohorts were scanned as WSI at 20x magnification using the Teksqray SQS-40P scanner, with a scanner resolution of 0.190 µm/pixel.

In the initial stage of data processing, we implemented a series of rigorous steps to ensure that our model could be trained and validated under optimal conditions.

First, we performed stratified sampling on all case data, dividing it into training (60%), validation (20%), and test cohorts (20%) to ensure that each set represented the characteristics of the overall dataset. Next, to remove interference from background and blurry areas, we applied red, green, blue (RGB) thresholding and canny edge detection to segment the tissue regions in the H&E-stained WSI images, accurately distinguishing white background and blurry areas in the images.

To mitigate the bias from data originating from different centers, we performed normalization as follows. We sliced the WSI images into $512 \times 512$-pixel patches, with each pixel at a resolution of 0.5 µm. After that, Z-Score normalization was applied to each patch, and the normalization formula is as follows:

$$y = \frac{x - mean(x)}{std(x)} \quad (1)$$

where $x$ is the value of a pixel in a certain channel of the input image, and the mean($\cdot$) and std($\cdot$) functions are used to calculated the mean and standard deviation values of the corresponding channels.

The data from different centers may differ in terms of the proportion of positive and negative samples, and the class weight in the binary cross-entropy loss function is used to balance the proportion of positive and negative samples. When calculating the loss, for each sample, the loss is multiplied by the corresponding class weight based on its class. In this way, during back propagation, the model adjusts the parameter update according to the class weight, ensuring that the learning of samples from different categories is balanced.

For biopsy samples, due to the limited sample area, we used a sliding window enhancement algorithm to increase the number of patches. We then applied data augmentations such as rotation and flipping to each patch to effectively expand the variety of patch images.

Finally, to address the imbalance between samples with many patches and those with fewer, we applied random undersampling to the former and random data augmentation to the latter, including operations such as rotation, flipping, scaling, cropping, blurring, Gaussian noise, and salt-and-pepper noise to equalize the patch count for each label. Considering the quadratic complexity of the self-attention mechanism and the large number of patches per WSI image, and given GPU memory limitations (24 GB), we capped the number of patches per label at 12,000.

This series of steps was designed to ensure that our model could fully learn the features within the dataset, thereby achieving strong generalization and improving its performance and accuracy in practical applications.

Local and cross-center validations were performed on the test set, an independent external test set, and the TCGA-TNBC dataset. The models' prediction performance was compared to confirm their generalizability. Due to the limited number of TNBC patients in the three self-collected datasets, we did not train or validate separately on this group, but instead conducted a statistical analysis of the predicted results. The PD-L1 labels for the TCGA-TNBC dataset were determined based on the study by *Jin et al. (2024)*, where RNA-FPKM values were divided at the 75th percentile; values above this threshold were labeled as PD-L1 positive, and values below this threshold were labeled as negative.

## Feature extration

We used the CTransPath model as our feature extraction network to capture a 768-dimensional feature vector for each patch. This model is based on the Swin Transformer architecture, which combines the hierarchical structure of CNNs with the global self-attention module of transformers by computing self-attention within a sliding window. The model starts with three convolutional layers to facilitate local feature extraction and enhance training stability, followed by four Swin Transformer stages. CTransPath was pre-trained on untokenized histopathological images using unsupervised contrastive loss with data from The Cancer Genoma Atlas (TCGA) and the Pathology Artificial Intelligence Platform (PAIP) across multiple organs. This enables the model to act as a collaborative local–global feature extractor, learning generalized feature representations better suited

to histopathological image tasks. Each patch's embedding is stored for the subsequent training process.

## Overview of the MIL model

The MIL network architecture for predicting PD-L1 expression in breast cancer histopathological images consists of three main modules: data preprocessing, feature extraction, and the MIL PD-L1 detection network.

First, histopathological images undergo normalized preprocessing to unify image quality and minimize bias due to different scanning conditions. Next, these patches are fed into the CTransPath-based feature extraction network (*Wang et al., 2022*). The CTransPath network automatically learns deep features within the images, transforming them into high-dimensional vector representations. This high-dimensional vector captures spatial, textural, and structural features in the pathology images, encoding microscopic variations in tumor regions and their surrounding environment.

The high-dimensional features are then aggregated and analyzed by a weakly supervised PD-L1 detection network based on MIL. The MIL framework learns the relationship between multiple patch features and PD-L1 expression, effectively handling information inconsistencies across patches, thereby enabling accurate PD-L1 expression prediction under weak supervision for each histopathological image. Finally, the model produces a PD-L1 prediction based on the aggregated features of multiple patches and outputs a confidence score to assess the reliability of the result.

### *MIL-based weakly supervised aggregation modules*

AttentionMIL is a classical MIL method, which can adaptively assign weights to each instance according to its importance to the global task by introducing an attention mechanism (*Ilse, Tomczak & Welling, 2018*). Its advantage lies in the ability to effectively identify critical regions from large-scale WSI data, which is suitable for the task of biomarker detection that needs to focus on local features.

The aggregation module in the AttentionMIL model uses an attention mechanism to combine the features of each patch. By assigning a learnable attention weight to each patch, the model can automatically focus on more relevant patches, giving them higher weights. The final classification result is generated by weighted summation of all patch features according to their attention weights. This attention mechanism enables the model to autonomously identify critical regions, enhancing its ability to capture key biomarker information—especially useful in weakly supervised learning scenarios.

Clustering-constrained Attention Multiple Instance Learning (CLAM) can effectively improve the classification performance under weakly supervised conditions by introducing Clustering constraints and combining label information with instance-level features (*Lu et al., 2021*). It is designed to alleviate the problem of inconsistent instance labels in multi-instance learning and has high robustness for the classification of specific regions in breast cancer PD-L1 detection.

The aggregation module in the CLAM model introduces an instance selection mechanism that not only focuses on key patches but also explicitly filters out patches with important

features for the classification task. CLAM first generates each patch's features through a pre-trained feature extraction module, then further selects the most crucial patch features for target classification using the instance selection and weighted aggregation module. Unlike pure attention-based aggregation, CLAM's instance selection mechanism enables the model to concentrate on more discriminative local areas, making it more effective in weakly supervised learning with noisy data.

TransMIL, which based on the Transformer architecture, TransMIL is able to capture more complex global context information and long-range dependencies between instances (*Shao et al., 2021*). This feature makes it have a significant advantage in dealing with WSI data with both global and local features, especially in the detection task of PD-L1 expression that requires comprehensive information of the whole film. Since TransMIL is based on a Transformer architecture, it is good at capturing global context information and long-range dependencies between instances. In the task of PD-L1 detection in breast cancer, PD-L1 expression often needs to integrate the global features of the whole film and the important information of the local region. Compared with AttentionMIL and CLAM, TransMIL is more suitable to deal with such complex tasks that require the cooperation of global and local features, and shows better detection results.

The TransMIL model's aggregation module is based on the Transformer architecture and aggregates patch features through a self-attention mechanism, allowing all patch features to interact and establish long-distance dependencies between features. The self-attention weights for each patch are computed using a multi-head attention mechanism, effectively capturing complex inter-patch relationships. Additionally, TransMIL incorporates the Pyramid Position Encoding Generator (PPEG) as a positional encoding module, providing multi-scale positional encoding that enhances the model's integration of global and local information, resulting in improved generalizability across different data centers. During aggregation, the model dynamically identifies and emphasizes globally significant and highly relevant features. This Transformer-based aggregation approach not only improves the model's ability to capture complex biomarker information but also enables classification decisions from a global perspective, resulting in superior robustness and generalization across diverse datasets.

## Heatmap visualization of interpretable features

The heatmap is generated by extracting the image patch features in the WSI and combining the model-generated attention scores to map to the original coordinates of the WSI. The scores can be normalized to percentiles for comparison, resulting in a visual heat map that covers the entire WSI and visually shows the model focus areas and their importance. The MIL-based approach enables the generation of highly interpretable heatmaps. During model training, pixel-level annotations are not required, yet the model can intuitively display the relative contribution and importance of each tissue area to its predictions, facilitating clinical analysis. We used a feature aggregation model that assigns an attention score to each patch.

In the heatmap, higher attention values (shown in red) indicate areas that contribute more significantly to the model's prediction, while lower attention values (shown in blue)

represent less impactful regions. By mapping these attention values across the whole image, the resulting heatmap clearly highlights the regions the model focuses on most.

This heatmap is a valuable tool for helping clinicians understand the model's decision-making process. Red areas suggest regions the model considers more likely to be associated with lesions or critical pathological features, whereas blue areas are seen as less relevant to the prediction. This visualization not only enhances model interpretability but also provides meaningful guidance for clinical diagnosis.

## Statistical analysis

The model classification results are visualized as a confusion matrix. According to the confusion matrix, we use area under curve (AUC), accuracy (ACC), precision (PRE), specificity (SPE), sensitivity (SEN) and F1-score as our statistical metrics to quantify the performance of the method. AUC with 95% confidence intervals (CI) were calculated to evaluate the stability and accuracy of these models. The specific formulas are shown as Eqs. (2)–(6).

$$ACC = \frac{TP + TN}{TP + TN + FP + FN} \tag{2}$$

$$SEN = \frac{TP}{TP + FN} \tag{3}$$

$$SPE = \frac{TN}{TN + FP} \tag{4}$$

$$PRE = \frac{TP}{TP + FP} \tag{5}$$

$$F\text{-}1 = \frac{2 \times SEN \times PRE}{SEN + PRE} \tag{6}$$

where TP = true positive, TN = true negative, FP = false positive, and FN = false negative.

# RESULTS

## Experimental setup

Experiments were conducted using Python 3.8 and PyTorch on a server equipped with a single NVIDIA 4090 GPU and an Intel Core i9-14900K CPU. This hardware configuration and software environment ensured efficiency and consistency during the model training process and provided high-performance computing resources to support the complex computational requirements of the models.

The Adam optimizer was used for all three models training. The weight decay parameter of each model was 1e–2 for AttentionMIL, 1e–4 for CLAM and 1e–5 for TransMIL. While the learning rate was 5e–4 for AttentionMIL, 2e–4 for CLAM and 5e–4 for TransMIL.

**Table 2 AUC and 95% CI of each model in the two test sets.** Three models were evaluated on internal (NF+HY) and external (XZ) test sets. AUC and 95% CI are reported for each model.

| | NF+HY test | XZ test |
|---|---|---|
| AttentionMIL | 0.806 (95% CI [0.744–0.869]) | 0.778 (95% CI [0.726–0.825]) |
| CLAM | 0.815 (95% CI [0.753–0.872]) | 0.786 (95% CI [0.735–0.831]) |
| TransMIL | 0.833 (95% CI [0.768–0.892]) | 0.799 (95% CI [0.751–0.846]) |

All models were trained for a total of 30 epochs. Due to GPU memory limitations, the batch size was set to 8. In each epoch, we employed the MultiStepLR learning rate adjustment strategy, reducing the learning rate to 0.1 times the original value at the 15th and 25th epochs to enhance the convergence and stability of the models. This optimized setup was then applied to evaluate the models on the test set. In order to guarantee the accuracy of the experimental results, all experimental results are the average of three runs.

## Experimental results

We developed three deep learning models: AttentionMIL, CLAM, and TransMIL. We tested the ability of these three models to predict PD-L1 expression directly from pathology images using the NF and HY breast cancer datasets. Due to the limited size of the datasets, we divided them into 60% training, 20% validation, and 20% testing sets. The results showed that the TransMIL model achieved the highest AUC of 0.833 on the NF and HY test sets, outperforming CLAM (0.815) and AttentionMIL (0.806) (Figs. 2A–2D). The scatter plots visualized the classification of the results of each model on the test sets (Figs. 3A–3C). To further evaluate the stability and accuracy of AUC, the 95% CI of the three models in the two test sets were calculated (Table 2). The results showed that the 95% CI of the three models in the two test sets were all around 0.1, and the classification performance of the models was relatively stable. However, the AUC 95% CI of the three models had a large overlap, and the differences between the models were not significant. Therefore, we further calculated the accuracy, precision, sensitivity, specificity and F1-score of each model (Table 3). In the NF and HY test sets, AttentionMIL and CLAM performed similarly, but both had slightly lower accuracy and precision compared to TransMIL, while their sensitivity lagged behind TransMIL, indicating that TransMIL performed best in detecting positive samples (Fig. 3G).

Given the significant differences in images between surgical and biopsy samples, we used the best-performing TransMIL model to separately analyze the prediction results for both surgical and biopsy samples. The results showed that the mean AUC for surgical samples was 0.856, while the AUC for biopsy samples was 0.762 (Fig. 4B). This indicates that the model performs better, or at least as well, for surgical samples. Together, these results demonstrate that our models can predict PD-L1 expression levels from H&E images, with better performance for surgical samples than for biopsy samples.

## Generalization performance

To further validate the generalization ability of the models, we used an independent external validation set, the XZ breast cancer biopsy data, for testing. The data from the XZ cohort
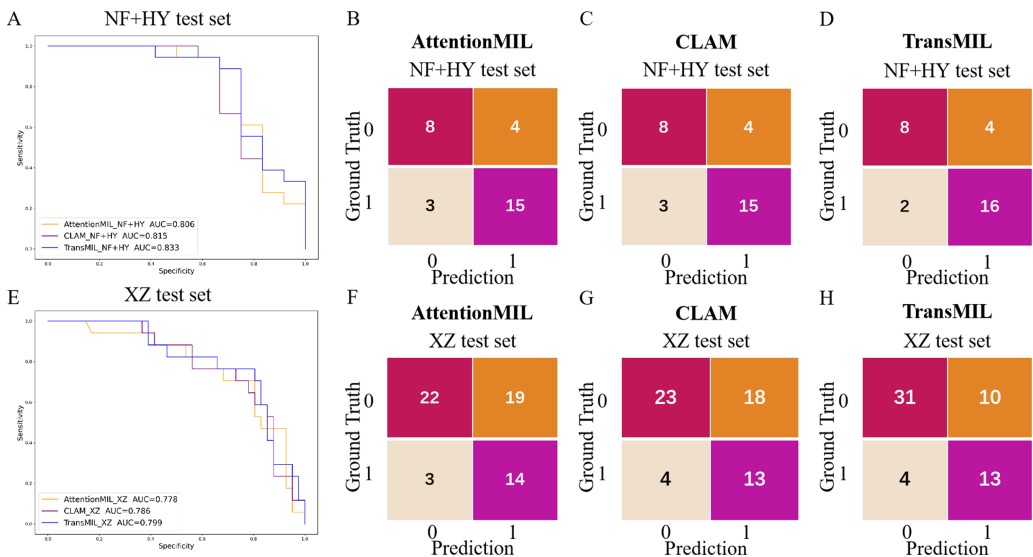

**Figure 2** **Performance overview of AttentionMIL, CLAM and TransMIL predicting the PD-L1 expression level in test sets.** (A) The receiver operating characteristic (ROC) curves for PD-L1 prediction by the three models in the NF+HY test set, represented by AUC. (B–D) The confusion matrices for PD-L1 prediction results from the three models in the NF+HY test set. (E) The receiver operating characteristic (ROC) curves for PD-L1 prediction by the three models in the XZ test set, represented by AUC. (F–H) The confusion matrices for PD-L1 prediction results from the three models in the XZ test set.

**Table 3** **Statistical metrics of each model in the two test sets.** Three models were evaluated on internal (NF+HY) and external (XZ) test sets by statistical metrics including AUC, ACC, PRE, SEN, SPE and F1-score.

|  | AUC | ACC | PRE | SEN | SPE | F1-score |
|---|---|---|---|---|---|---|
| AttentionMIL NF+HY test | 0.806 | 0.767 | 0.789 | 0.833 | 0.667 | 0.811 |
| CLAM NF+HY test | 0.815 | 0.767 | 0.789 | 0.833 | 0.667 | 0.811 |
| TransMIL NF+HY test | 0.833 | 0.800 | 0.800 | 0.889 | 0.667 | 0.842 |
| AttentionMIL XZ test | 0.778 | 0.621 | 0.424 | 0.824 | 0.537 | 0.560 |
| CLAM XZ test | 0.786 | 0.621 | 0.419 | 0.765 | 0.561 | 0.542 |
| TransMIL XZ test | 0.799 | 0.759 | 0.565 | 0.765 | 0.756 | 0.650 |

exhibits distinct characteristics compared to those of NF and HY datasets. The TransMIL model achieved an AUC of 0.799 on the XZ test set, showing a decrease of approximately 4.1% compared to the NF and HY test sets (Fig. 4A). CLAM and AttentionMIL models also showed a similar decline in performance on the XZ test set, with AUC values of 0.786 and 0.778, respectively (Figs. 2E–2H). The scatter plots were visually displayed the classification of the results of each model on the XZ test sets (Figs. 3D–3F). We also

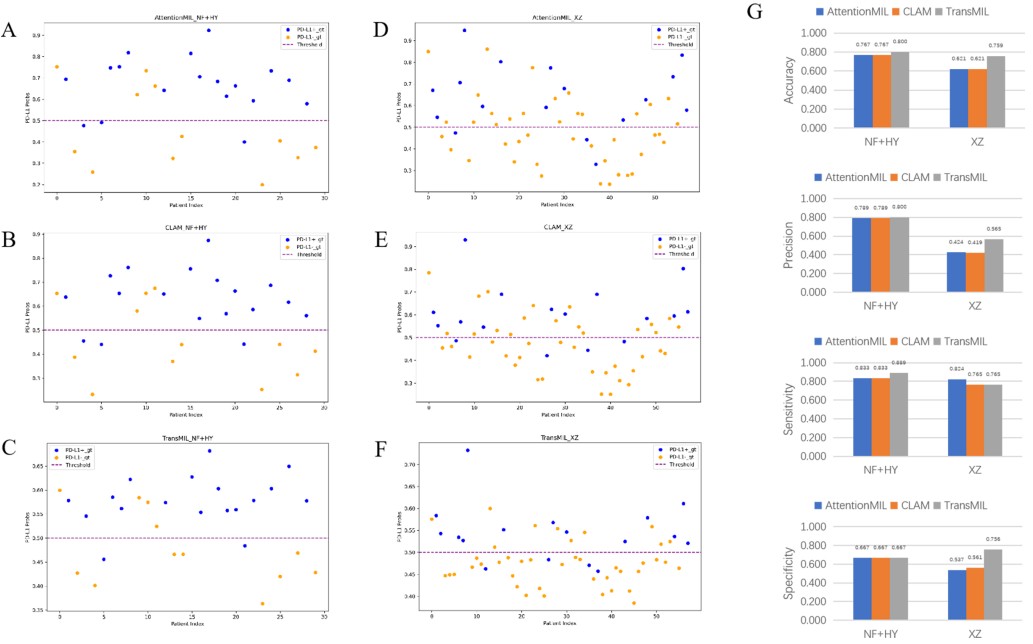

**Figure 3** **Statistical analysis of performance of AttentionMIL, CLAM and TransMIL predicting the PD-L1 expression level in test sets.** (A–C) Scatter plots of PD-L1 prediction results by the three models in the NF+HY test set. (D–F) Scatter plots of PD-L1 prediction results by the three models in the XZ test set. The $x$-axis represents the patient index, and the $y$-axis represents the predicted score for the WSI. The blue dashed line represents a threshold of 0.5. Patients above the dashed line are predicted as positive, and those below the dashed line are predicted as negative; blue points represent true positives, and orange points represent true negatives. (G) Bar plots of PD-L1 prediction results by the three models in the NF+HY and XZ test sets. Accuracy, precision, sensitivity, and specificity were statistically analyzed. Blue represents the AttentionMIL model, orange represents the CLAM model, and gray represents the TransMIL model.

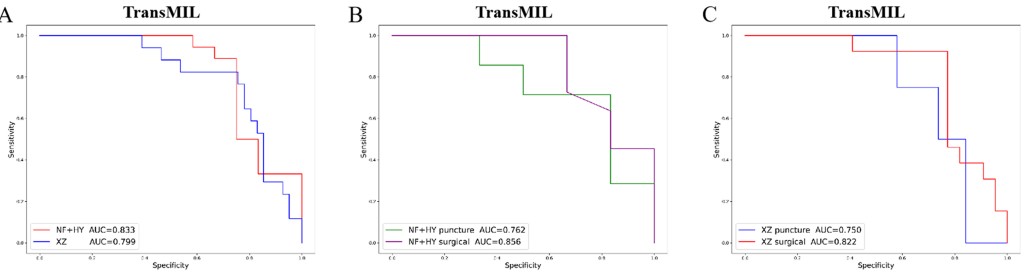

**Figure 4** **The overall performance of the TransMIL model in predicting PD-L1 expression in the NF+HY and XZ test sets, and the comparison of performance between surgical and biopsy samples.** (A) The receiver operating characteristic (ROC) curve of the TransMIL model predicting PD-L1 expression in the NF+HY and XZ test sets, represented by AUC. (B) The ROC curve of the TransMIL model predicting PD-L1 expression for surgical and biopsy samples separately in the NF+HY test set, represented by AUC. (C) The ROC curve of the TransMIL model predicting PD-L1 expression for surgical and biopsy samples separately in the XZ test set, represented by AUC.

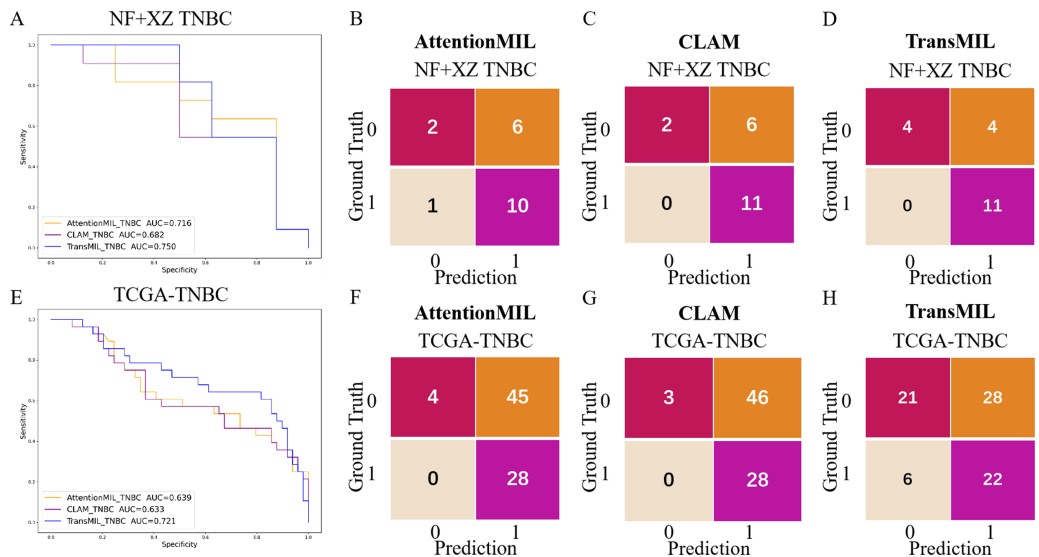

**Figure 5** **Performance overview of AttentionMIL, CLAM and TransMIL predicting the PD-L1 expression level in TNBC sets.** (A) The receiver operating characteristic (ROC) curve of the three models predicting PD-L1 in the NF+XZ TNBC dataset, represented by AUC. (B–D) The confusion matrices of the three models predicting PD-L1 results in the NF+XZ TNBC test set. (E) The receiver operating characteristic (ROC) curve of the three models predicting PD-L1 in the TCGA-TNBC test set, represented by AUC. (F–H) The confusion matrices of the three models predicting PD-L1 results in the TCGA-TNBC test set.

calculated the accuracy, precision, sensitivity, and specificity for each model (Fig. 3G). In terms of accuracy, precision, and specificity, TransMIL outperformed both AttentionMIL and CLAM, demonstrating superior classification ability. However, AttentionMIL achieved the highest sensitivity, indicating its superior performance in detecting positive samples in the XZ test set.

When the best-performing TransMIL model was applied to analyze the predictions for surgical and biopsy samples in the XZ test set, the results showed an AUC of 0.822 for surgical samples, compared to 0.750 for biopsy samples (Fig. 4C). These results indicate that all three models demonstrate good generalization ability on the independent external test set XZ, with TransMIL still performing the best, and surgical samples outperforming biopsy samples.

Studies have shown that using extensive RNA sequencing data can predict PD-L1 expression in H&E-stained slides. This suggests that RNA sequencing results have a threshold that allows H&E pathological images to be distinguished. Given the limited number of TNBC cases in this study (only 19 cases), the TransMIL model achieved an AUC of 0.750, while AttentionMIL and CLAM performed lower, with AUC values of 0.716 and 0.682, respectively (Figs. 5A–5D). Therefore, we aimed to use TCGA TNBC data to validate the consistency between our models and RNA-based prediction models.

Based on previous study (*Jin et al., 2024*), we selected TNBC cases from the TCGA cohort and set the RNA FPKM value threshold as the 75th percentile, which was 1.9355. Values above this threshold were classified as positive, while those below were classified as

A

**TransMIL_TCGA-TNBC**

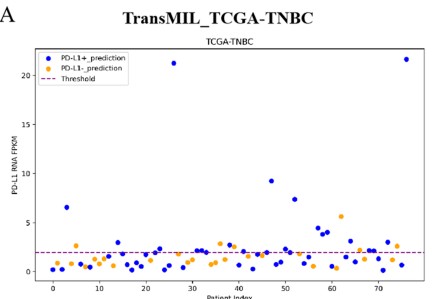

B

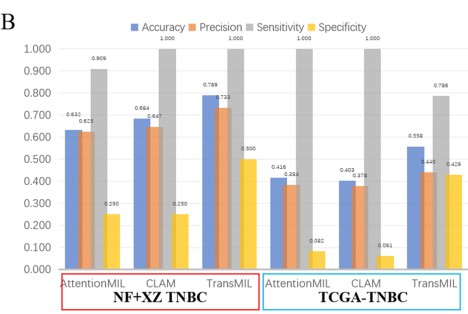

**Figure 6 Statistical analysis of performance of AttentionMIL, CLAM and TransMIL predicting the PD-L1 expression level in TNBC sets.** (A) The scatter plot of the three models predicting PD-L1 results in the TCGA-TNBC test set. The *x*-axis represents the patient index, and the *y*-axis represents the corresponding patient's RNA FPKM value. The blue dashed line is the threshold, which in this study is set as the upper quartile of the RNA FPKM value, *i.e.,* 1.9355. Patients above the dashed line are considered positive based on the RNA FPKM value, while those below the line are considered negative. Blue dots represent patients predicted as positive by the model, and orange dots represent patients predicted as negative by the model. (B) The bar chart of the three models predicting PD-L1 results in the NF+XZ TNBC and TCGA-TNBC test sets. Statistical analysis was performed for accuracy, precision, sensitivity, and specificity. The three sets of data within the red box represent the NF+XZ TNBC test set, while the three sets of data within the blue box represent the TCGA-TNBC test set.

negative. In the TCGA TNBC dataset, the TransMIL model achieved an AUC of 0.721, while CLAM and AttentionMIL performed lower, with AUCs of 0.639 and 0.633, respectively (Figs. 5E–5H). The prediction performance of all three models in the TCGA-TNBC dataset was lower compared to the NF and XZ datasets (Figs. 5A, 5E).

A scatter plot of the prediction results from the best-performing TransMIL model revealed that the model had very few false negatives but a relatively high number of false positives (Fig. 6A). We also calculated accuracy, precision, sensitivity, and specificity for the predictions of the NF and XZ_TNBC and TCGA-TNBC datasets. The results showed that all three models had high sensitivity, even reaching 1, but their specificity was relatively low, especially for AttentionMIL and CLAM in the TCGA-TNBC dataset (Fig. 6B).

Although the high sensitivity (close to 1) indicates that the models performed well in detecting positive cases, the low specificity, particularly for TransMIL which performed below 0.5, reflects a high false-positive rate. While this may reduce the chance of missing positive cases in clinical practice, it could also lead to an increased economic burden and time costs for a significant number of false-positive patients. Overall, we believe that the TransMIL model still performed well in predicting PD-L1 expression in TCGA-TNBC H&E slides, but further optimization and adjustments are needed for broader clinical application.

## DISCUSSION

### Comparison of performance differences between models

Multiple studies have proven that PD-L1 is an important biomarker for immune therapy response, with significant implications across various types of cancer (*Chen & Mellman, 2013*; *Upadhaya et al., 2022*; *Yi et al., 2022*). The FDA-approved PD-1/PD-L1

immunotherapies currently include treatments for lung cancer, colorectal cancer, TNBC, and other malignancies (*Lee et al., 2020*; *Nobin et al., 2024*; *Savic et al., 2019*). In the rapidly developing field of digital pathology, many tools for predicting molecular markers from tumor pathology slides have been developed (*Baxi et al., 2022*; *Bera et al., 2019*; *Shafi & Parwani, 2023*). In this study, we trained and made prediction using three models: AttentionMIL, CLAM, and TransMIL.

Whether on the NF and HY test sets or the independent XZ test set, the ACC and F1-scores of TransMIL were higher than those of CLAM and AttentionMIL, indicating its overall better classification performance. Sensitivity and specificity measure how well positive and negative examples were identified, respectively. From the experimental results, in the NF and HY test set, TransMIL had the highest sensitivity, but in, but in the XZ test set, TransMIL had the same sensitivity as CLAM and slightly lower sensitivity than AttentionMIL. In terms of specificity, the three models performed consistently on the NF and HY test set, but TransMIL was significantly higher than the other two models on the XZ test set. The sensitivity was high, indicating that the model performs better in detecting positive samples, but AttentionMIL and CLAM had significantly lower specificity, reflecting that their good detection of positive samples might be at the expense of detecting negative samples, and the model performance had a significant deviation. The accuracy also clearly reflects this feature, and in the NF and HY and XZ test sets, especially in the latter, TransMIL showed better results, demonstrating its better generalization ability.

For such differences, researchers believe that, on the one hand, it may be related to the characteristics of the Transformer architecture. The Transformer architecture in TransMIL better utilizes the feature relationships between different patches, capturing global information about biomarkers within tissue samples, making it more suitable for the label generation methods used in NF and HY. On the other hand, the Transformer-based long-distance feature aggregation mechanism helps the model better adapt to the data characteristics of Center 3 to a certain extent. In contrast, CLAMMIL's instance selection mechanism achieves a slightly lower AUC on Center 3, indicating that the instance selection strategy may be sensitive to more heterogeneous data when sample labels are generated in a way that deviates from the training data. On the other hand, AttentionMIL only uses attention weights, which makes it difficult to fully adapt to changes in data distribution and label generation methods, so it has relatively weak performance in cross-center testing.

Meanwhile, compared to the pre-trained NF and HY datasets, all models showed decreased performance on the independent external XZ test set. This may be due to differences in how labels were obtained. In NF and HY, PD-L1 labels were acquired individually from tissue samples using conventional full-slide immunohistochemistry methods, whereas XZ used a tissue microarray (TMA) method for immunohistochemical detection. The tissue microarray method requires high selectivity and tissue homogeneity, which may not fully represent the entire tissue sample, potentially leading to information loss. This could explain the performance decline of the models on the XZ test set. In addition, despite data normalization, there are differences in staining instruments and reagents across different centers, differences in data storage formats, and sample bias

caused by variations in race and living environment among datasets, which may lead to the performance degradation of each model on the XZ test set.

## Comparison of differences between datasets

This study involves data from three centers, primarily using formalin-fixed paraffin-embedded (FFPE) samples, including both biopsy and surgical samples. This provides a solid sample foundation for exploring the stability of model performance across multiple centers and sample types. As an independent external validation set, the XZ dataset comes from the Xizang Autonomous Region in China, which is relatively underdeveloped compared to the economically advanced coastal regions. Medical resources in the region are limited, and maintaining consistent quality in pathological slides is challenging. Additionally, the cohort is predominantly composed of the local Tibetan ethnic group. Given the unique high-altitude environment, cold climate, and distinct lifestyle, dietary habits, and other factors, the XZ cohort holds significant importance.

The experimental results show that the overall predictive AUC reaches around 0.800, demonstrating good generalization, with surgical samples showing better predictive accuracy than biopsy samples. On the one hand, surgical samples contain more information, including not only tumor regions but also tumor invasion edges, tumor-associated lymphocyte infiltration, and peritumoral stroma. On the other hand, biopsy samples are more technically demanding for the operator, requiring accuracy in the biopsy site, and are limited in size, potentially missing important surrounding tissue information. These factors may contribute to the observed performance differences between surgical and biopsy samples. Analyzing the locations of the three centers used in this study reveals that Center NF and Center HY are both located in the economically developed southern coastal regions, where sample collection is relatively more abundant, and the quality control during processes like sampling, preparation, and staining is stricter, resulting in more consistent samples. In contrast, Xizang is located in the inland, economically underdeveloped region with limited medical resources, and quality control in pathological slide preparation is relatively weaker. Despite image preprocessing steps like staining normalization, these disparities may still lead to differences in model performance.

## Visual heatmaps showing interpretability features

By visualizing heatmaps, we further explored the connection between model predictions and histological features. In molecular marker prediction tasks, heatmaps can highlight tissue regions that are highly correlated with specific molecular markers (*e.g.*, PD-L1 in this study) and analyze the relationship between tissue morphology and molecular marker expression, enabling pathologists to visualize the basis of the model's predictions and helping to understand how pathological features can influence changes at the molecular level. The attention heat map reveals the tissue structures that the model focuses on when making predictions, enhancing the transparency and interpretability of the model. In addition to regions of interest, regions of uncertainty and their confidence levels are also visualized, thus assisting pathologists in their review and reducing the risk of misclassification.

Previous studies have shown that features such as high nuclear heterogeneity, tumor necrosis, lymphovascular invasion, inflammatory cell infiltration, stromal reaction, tumor necrosis, and high-level tissue patterns are important in the diagnostic nuclear prognostic assessment of solid tumors and often correlate with poor tumor prognosis (*Arpinati & Scherz-Shouval, 2023*; *Bilotta, Antignani & Fitzgerald, 2022*; *Stanton & Disis, 2016*). In this study, the best predictive TransMIL model was subjected to heatmap visualization to observe the histological features of the focal patches to obtain clinical interpretability (Figs. 7A–7D). In the heatmap visualization results of TransMIL, the patches with high predictive scores exhibited features such as tumor cell nuclei with large and pronounced nucleoli, lymphocyte infiltration, and high-grade histological patterns (solid or micropapillary structures), which were consistent with previous study results. The features of large nuclei and pronounced nucleoli of tumor cells tend to be associated with features such as high proliferative activity, high malignancy, and genetic mutations, which in breast cancer often also suggests a possible association with chemotherapy tolerance. Tumor infiltrating lymphocytes play an important role in the prognostic assessment of breast cancer, and their effect on tumor is associated with distribution location, cell subtype, and tumor subtype. The model focuses on the characteristics of lymphocyte infiltration, which, while suggesting the site of abnormality, can also help clinicians to predict the response to treatment in conjunction with tumor typing. In addition, the high-level histological pattern focused on the heatmap was classified as the relatively worst group in terms of prognosis among the high, medium and low histological patterns classified by the World Health Organization (WHO) as recently as 2015 (*Moreira et al., 2020*; *Travis et al., 2015*). All of the above histological features suggest that the TransMIL model, which was the most predictive in this study, facilitates focusing on the lesion site and also allows for the acquisition of histological features associated with a poor prognosis, such as high-level histological patterns. The acquisition of such histological features allows the model to have more accurate predictive performance while providing clinicians with interpretable features for diagnosis and prognosis.

## Model performance in TNBC subgroup

Currently, clinically approved PD-L1 testing and corresponding treatments are primarily targeted at TNBC patients in breast cancer, while first-line treatments for non-TNBC patients still involve hormone therapy or Her-2 targeted therapy (*Emens & Loi, 2023*; *Prat et al., 2015*). To explore the predictive effectiveness of our model for TNBC patients, we classified patients based on ER, PR, and HER-2 information, and made predictions for a total of 19 patients from the NF and XZ centers. The results showed that the TransMIL model performed the best. The confusion matrix clearly indicates that all three models tend to predict TNBC patients as positive. In clinical applications, this would significantly reduce the rate of missed diagnoses. Since our TNBC sample size is very limited, it is difficult to determine how the model would perform on a larger scale. However, studies have shown that RNA sequencing can predict PD-L1 expression in H&E slides. Therefore, we also made predictions on the publicly available TCGA-TNBC dataset, and the results showed an AUC of 0.721. These results suggest that, not only in the TNBC dataset but

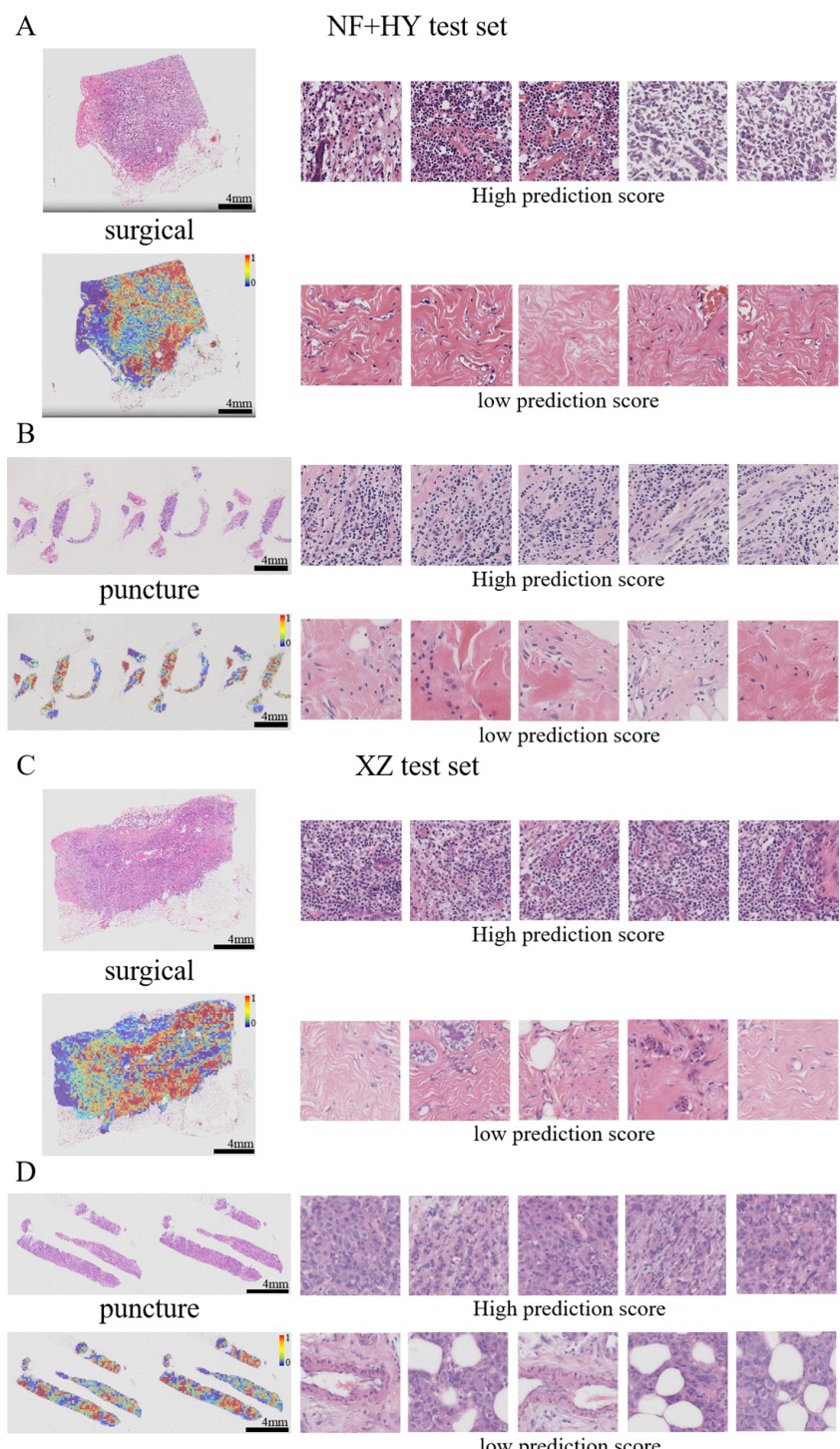

**Figure 7  Typical patterns for PDL1 high/low prediction in H&E slide images of test sets.** (A) H&E images, heatmaps, and the top five patches with the highest/lowest prediction scores for surgical and biopsy samples in the NF+HY test set. (B) H&E images, heatmaps, and the top five patches with the highest/lowest prediction scores for surgical and biopsy samples in the XZ test set. Tiles of high expression are marked in red, and low expression ones are marked in blue. Scale bars in the WSI views are all four mm.

also across all datasets, our model tends to identify patients as positive. We hypothesize the following reasons for this. First, the distribution of the test set differs from that of the training set, where the training set has more positive samples than negative ones, while both the independent external test set (XZ) and the TCGA-TNBC dataset have more negative samples than positive ones. The uneven sample distribution between the training and test sets, along with these discrepancies, may cause the model to lean towards predicting positivity. Secondly, although data from three centers were included, the sample size from each center is very limited, which may restrict the model's performance. Additionally, models trained on RNA sequencing results have shown an AUC of only 0.540 when predicting PD-L1 expression in FFPE pathological slides from TNBC. Even with better performance using fresh frozen tissue slides, the AUC is only 0.640. This suggests that while there is a morphological-transcriptomic link in PD-L1 expression, its separability in TNBC is generally limited. In the future, it is hoped that using more integrated morphological-transcriptomic data for training will improve the model's predictive performance.

## Innovations and limitations

In summary, this study combines multi-instance learning with the XZ dataset for integrated innovation. The MIL approach allows the AI to automatically learn key regions from WSI, freeing it from the dependence on pixel-level annotation, and improves the model's generalization ability while effectively reducing the workload of manual annotation. The XZ dataset is from the Tibetan region, where medical resources are comparatively limited and the cohort is predominantly composed of ethnic minorities. This results in disparities among cohort populations and in the level of preparation and reading of pathology slides. The model's reliable cross-center capability enables the implementation of training using data that is readily available in developed regions and subsequent prediction of data from less developed regions. This is important for achieving the efficient allocation of healthcare resources in regions with unbalanced economic development.

However, this paper also has some limitations. The first is the issue of data imbalance. Due to the limited size of the dataset, PD-L1 expression is unevenly distributed across the datasets, especially the TNBC subgroup, which shows opposite distribution characteristics in the training set and the independent external test set. Secondly, three local datasets were used in this study, with data originating from different pathology laboratories, each with differences in tissue processing methods, staining techniques, and technical practices of personnel. Despite data normalization, heterogeneity still existed among the data. Moreover, it is clear from the results that the performance of all three models on the XZ dataset is degraded, suggesting that race-specific and region-specific datasets may have an impact on the generalizability of the models. Finally, although the model achieved AUC up to about 0.800 in the local dataset, it still lacks clinical validation tests in a really patient population. In the future, we will increase the data size, address data imbalance and improve the normalization process, increase the model generalizability and deploy it in the clinic, applying and improving the model in real environments with a view to its early use.

## CONSLUSIONS

In summary, the TransMIL model based on the Transformer architecture achieves good performance in predicting PD-L1 expression in H&E slides using a multi-instance learning approach. The model's reliable cross-center capability allows us to train on data from well-developed regions, and then apply it to predict data from less developed regions. This has significant implications for the effective allocation of medical resources in economically imbalanced regions. The model's visualized heatmaps provide clinicians with interpretable features that link the model's predictions to clinical outcomes. These features are not only associated with PD-L1 expression levels but may also be relevant to clinical prognosis. In the future, increasing the data size and the number of centers is expected to further enhance the model's predictive performance. Additionally, incorporating diagnostic and treatment-related information to build a multimodal model could not only predict PD-L1 expression levels but also provide corresponding treatment recommendations.

### Funding

The present study was supported by the Xizang Natural Science Foundation (Grant No. XZ2022ZR-ZY08(Z)) and the 2022 Major Science and Technology Innovation R&D Project of Chongqing Municipality (project number: CSTB2022TIAD-STX0008). The funders had no role in study design, data collection and analysis, decision to publish, or preparation of the manuscript.

### Grant Disclosures

The following grant information was disclosed by the authors:
Xizang Natural Science Foundation: XZ2022ZR-ZY08(Z).
2022 Major Science and Technology Innovation R&D Project of Chongqing Municipality: CSTB2022TIAD-STX0008.

### Competing Interests

Heng Yang is an employee of Jinfeng Laboratory, and Kunyuan Fangqing Medical Technology Co., LTD.

Kaixiang Sun is an employee of Jinfeng Laboratory, and Kunyuan Fangqing Medical Technology Co., LTD.

Feiyan Wang is an employee of Jinfeng Laboratory, and Kunyuan Fangqing Medical Technology Co., LTD.

Weijun Pan is an employee of Jinfeng Laboratory, and Nanfang Hospital and Basic Medical College, Southern Medical University.

### Author Contributions

- Zhen Da conceived and designed the experiments, prepared figures and/or tables, authored or reviewed drafts of the article, and approved the final draft.

- Heng Yang performed the experiments, analyzed the data, prepared figures and/or tables, authored or reviewed drafts of the article, and approved the final draft.
- Bianba Zhaxi performed the experiments, prepared figures and/or tables, and approved the final draft.
- Kaixiang Sun performed the experiments, analyzed the data, prepared figures and/or tables, and approved the final draft.
- Guohui Bai conceived and designed the experiments, prepared figures and/or tables, and approved the final draft.
- Chao Wang performed the experiments, analyzed the data, prepared figures and/or tables, and approved the final draft.
- Feiyan Wang performed the experiments, prepared figures and/or tables, and approved the final draft.
- Weijun Pan conceived and designed the experiments, authored or reviewed drafts of the article, and approved the final draft.
- Rui Du conceived and designed the experiments, authored or reviewed drafts of the article, and approved the final draft.

## Human Ethics

The following information was supplied relating to ethical approvals (i.e., approving body and any reference numbers):

The study was approved by the Ethics Committee of People's Hospital of Xizang Autonomous Region (No. ME-TBHP-22-KJ-041) on July 22, 2023; and a waiver of informed consent was submitted to the same committee on July 4, 2022.

## Data Availability

The data is available at figshare: Du, Rui (2024). data.zip. figshare. Dataset. https://doi.org/10.6084/m9.figshare.27898149.v3.

The data is available at GitHub and Zenodo:

- https://github.com/sun-kx/PD-L1-expression-prediction

- sun-kx. (2025). sun-kx/PD-L1-expression-prediction: PD-L1-expression-prediction (v1.0.0). Zenodo. https://doi.org/10.5281/zenodo.15043037.

## Supplemental Information

Supplemental information for this article can be found online at http://dx.doi.org/10.7717/peerj.19201#supplemental-information.

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
