# Peer review of "Multiple instance learning-based prediction of programmed death-ligand 1 (PD-L1) expression from hematoxylin and eosin (H&E)-stained histopathological images in breast cancer"

_PeerJ, doi:10.7717/peerj.19201_

## Round 0.1 · original submission · Major Revisions

Based on the reviews received from two referees, while their recommendations differ between major and minor revisions, I have carefully evaluated their detailed comments and decided that major revision is required for your manuscript. Please address all the reviewers' comments thoroughly and submit a point-by-point response along with your revised manuscript. Pay particular attention to the major concerns raised by Reviewer #2. We look forward to receiving your revised version.

Reviewer 1 ·

Basic reporting

This study incorporated data from multiple centers to verify the potential of the multiple instance learning method in detecting the PDL-1 biomarker, which has certain application prospects. This work provides a valuable reference for rational and efficient allocation of medical resources. However, I think the article still needs further improvement.
Here are some major comments:
1) The structure of this paper is slightly complex. For example, “3. Data Preprocessing” in “Methods” section and “1. Model Workflow” in the “Results” section are repetitive. Also, there are seven subheadings the “Results” section, which is too many. It is recommended that the author reorganize the article with reference to the following structure: “
2. Materials and Methods
2.1 Data Acquisition and Processing
2.2 Overview of the MIL model
2.2.1 Feature Extraction
2.2.2 MIL-Based Weakly Supervised Aggregation Modules
3. Results
3.1 Experimental setup
3.2 Experimental results
3.3 Generalization Performance
4. Discussion”
2) The citation of references in the article is not standardized. The author needs to carefully check according to the requirements of the journal and academic papers.
3) In the second paragraph of the introduction, the full name of “FDA” is not shown, Authors should check throughout the text for acronyms that appear for the first time and indicate the full name.
4) In the fourth paragraph of the introduction, the “traditional methods” need to be accurately described to avoid confusion with traditional machine learning algorithms.
5)The “Discussion” section consists of only one paragraph, which is overly long. Moreover, the heatmap visualization would be more appropriately placed in the “Discussion” section as the “Results” section is more concerned with presenting the outcomes. And the title “5. Visualized Heatmaps Provide Interpretable Features for the Process of Predicting Biomarkers from Histology and Their Correlation with Clinical Diagnosis and Treatment” is too long.
6)The author should emphasize the innovative contribution of this work. The limitations of the article should also be included in the discussion section.
7) Authors should unify the representation of fractions, using either decimals or percentages, rather than both.
8)Good English expression can effectively help the author to better understand the author's intention, and the author should ask a professional who is good at English to improve the quality of writing throughout the article.

Experimental design

Merge in Basic reporting

Validity of the findings

Merge in Basic reporting

Reviewer 2 ·

Basic reporting

PD-L1 is a crucial biomarker increasingly used in breast cancer immunotherapy, but current immunohistochemical quantification methods are time-consuming, expensive, and unreliable. H&E staining, a standard procedure in cancer pathology, is easily accessible and highly reliable. This study investigates the use of a weakly supervised multiple instance learning (MIL) approach, leveraging deep learning techniques, to predict PD-L1 expression from hematoxylin and eosin (H&E) stained images in breast cancer. The study utilizes a Transformer-based TransMIL model that effectively captures highly heterogeneous features within the MIL framework, achieving an area under curve (AUC) of 0.833 on an internal test set, 0.799 on an independent external test set, and 0.721 on the public TCGA-TNBC database. These results indicate strong cross-center generalization ability for the model. The findings highlight the potential of appropriate deep learning techniques to effectively predict PD-L1 expression from limited datasets, across various regions and centers. This advancement not only demonstrates the significant potential of deep learning in pathological AI but also provides valuable insights for rational and efficient allocation of medical resources. However, there are still many problems in this study that need to be solved or improved.

Experimental design

1. The study focuses solely on model performance on benchmark datasets, with no mention of clinical validation. Evaluating the model's performance in a real-world clinical setting, with patient-specific data and expert input, is essential to assess its practical applicability and clinical utility.
2. While the introduction mentions the limitations of IHC for PD-L1 assessment, it doesn't explicitly state the problem that this study addresses. Clearly stating the need for an alternative approach, particularly one that leverages H&E-stained images, would make the introduction more compelling.
3. The methods describes the feature extraction network (CTransPath) and the three aggregation modules (AttentionMIL, CLAM, and TransMIL). However, it lacks context regarding why these specific models were chosen. The authors should provide a brief rationale for each model selection, explaining why these models are suitable for the task and how they address potential challenges.

Validity of the findings

4. The methods briefly mentions the use of heatmaps for interpretability. However, it doesn't clearly explain how these heatmaps are generated or what specific features they highlight. Providing more detail on the heatmap generation process and the interpretation of the results would enhance the section's clarity.
5. The methods section acknowledges the use of data from different centers. The author should explain the steps taken to mitigate these biases, such as standardization or data normalization.
6.  The results in Figure 2 mention that TransMIL outperforms the other models but don't provide statistically significant comparisons. For example, a p-value or confidence interval for the difference in AUC between TransMIL and the other models would strengthen the claim of its superior performance.
7. The results primarily focus on AUC, while other important metrics like sensitivity, specificity, and accuracy are only mentioned briefly. A more comprehensive analysis of these metrics would provide a better understanding of model performance. In addition, while AUC is a useful metric, it doesn't tell the whole story. For example, a high AUC could be misleading if the model has low sensitivity, meaning it misses many positive cases.
8. The results mention the use of heatmaps to identify regions associated with PD-L1 expression, but they fail to connect these findings to clinical relevance. How do these identified features contribute to the diagnosis or treatment of breast cancer?
9. The discussion primarily focuses on TransMIL's performance while neglecting a more comprehensive analysis of the strengths and weaknesses of all three models. The discussion attributes the performance decrease on the XZ dataset to variations in the labeling methods. While this is a plausible explanation, further investigation into the specific differences in IHC protocols and tissue characteristics between the datasets is needed.
10. Provide a more detailed and nuanced explanation for the performance differences observed across datasets, considering potential factors such as variations in staining protocols, tissue characteristics, and labeling methods. Discuss the limitations of the study, including data imbalance, data heterogeneity, generalizability, and clinical validation.

Additional comments

11. The images require improvements in resolution, quality, and captions. Additionally, grammar polishing is needed to enhance readability.

---

## Round 0.2 · accepted · Accept

The reviewers have acknowledged that you have adequately addressed all previous concerns, and the revised manuscript now meets our journal's standards. The editorial office will contact you shortly regarding the next steps in the publication process.

Reviewer 1 ·

Basic reporting

The authors' revisions and responses are comprehensive and there are no more comments.

Experimental design

The authors' revisions and responses are comprehensive and there are no more comments.

Validity of the findings

The authors' revisions and responses are comprehensive and there are no more comments.

Reviewer 2 ·

Basic reporting

No further comment

Experimental design

No further comment

Validity of the findings

No further comment